# Transcriptome Analysis of the Responses of Rice Leaves to Chilling and Subsequent Recovery

**DOI:** 10.3390/ijms231810739

**Published:** 2022-09-15

**Authors:** Zhong Li, Muhammad Umar Khan, Puleng Letuma, Yuebin Xie, Wenshan Zhan, Wei Wang, Yuhang Jiang, Wenxiong Lin, Zhixing Zhang

**Affiliations:** 1Fujian Provincial Key Laboratory of Agroecological Processing and Safety Monitoring, College of Life Sciences, Fujian Agriculture and Forestry University, Fuzhou 350002, China; 2Key Laboratory of Crop Ecology and Molecular Physiology, Fujian Agriculture and Forestry University, Fuzhou 350002, China; 3Crop Science Department, Faculty of Agriculture, National University of Lesotho, Maseru 100, Lesotho; 4Agricultural College, Fujian Agriculture and Forestry University, Fuzhou 350002, China

**Keywords:** temperature variation, *Oryza sativa* L., transcriptional regulation, ion balance, growth and development, adversity stress, energy metabolism, level of stress

## Abstract

Improving chilling tolerance at the seedling stage in rice is essential for agricultural research. We combined a physiological analysis with transcriptomics in a variety Dular subjected to chilling followed by recovery at normal temperature to better understand the chilling tolerance mechanisms of rice. Chilling inhibited the synthesis of chlorophyll and non-structural carbohydrate (NSC) and disrupted the ion balance of the plant, resulting in the impaired function of rice leaves. The recovery treatment can effectively reverse the chilling-related injury. Transcriptome results displayed that 21,970 genes were identified at three different temperatures, and 11,732 genes were differentially expressed. According to KEGG analysis, functional categories for differentially expressed genes (DEGs) mainly included ribosome (8.72%), photosynthesis–antenna proteins (7.38%), phenylpropanoid biosynthesis (11.41%), and linoleic acid metabolism (10.07%). The subcellular localization demonstrated that most proteins were located in the chloroplasts (29.30%), cytosol (10.19%), and nucleus (10.19%). We proposed that some genes involved in photosynthesis, ribosome, phenylpropanoid biosynthesis, and linoleic acid metabolism may play key roles in enhancing rice adaptation to chilling stress and their recovery capacity. These findings provide a foundation for future research into rice chilling tolerance mechanisms.

## 1. Introduction

Temperature is one of the leading environmental factors affecting plant distribution in terrestrial ecosystems [1]. Therefore, crop losses caused by chilling are an important worldwide problem. The plant damage caused by chilling has become more serious with the continuous deterioration of the global ecological environment. Chilling weakens the cell membrane, resulting in reactive oxygen species (ROS) and other harmful substances, and impacts plant vegetative growth and yield formation [2,3,4]. Several studies have focused on the molecular response of rice to chilling and its physiological and ecological characteristics [5,6,7]. Studies have demonstrated that the growth index and the physiological enzyme activity of rice are significantly inhibited under chilling, although this effect becomes smaller after the chilling pretreatment of rice [8]. This finding indicated that the chilling tolerance of rice could be induced. Because of this, studies have found that chilling environments can induce the expression of multiple genes [9,10,11,12]. The proteins encoded by these genes are involved in the sensing and signal transduction processes of plant tolerance to chilling to promote the synthesis of plant osmotic regulators [13]. They also enhance the activity of antioxidant enzymes [14] and improve cell membrane fluidity [15], thus enhancing the plant’s ability to withstand chilling [16,17]. Therefore, revealing the process and mechanism of sensing, transmitting chilling signals, and stimulating chilling tolerance of plants is of great theoretical and practical significance for further understanding the chilling tolerance mechanism of plants and accelerating the molecular breeding process in crops with chilling tolerance.

Existing studies have revealed that the transcriptional regulation of chilling response genes is an important process in plant responses to chilling injury [18]. Among these, the signal dependent on CBF/DREB, is an essential signaling pathway for *Arabidopsis thaliana* and rice to produce chilling tolerance ability [7]. The calcium signaling pathway is activated after the plant accumulates chilling tolerance characteristics, and the genes involved are evolutionarily conserved [16]. Through in-depth research, Ma et al. found that *COLD1*, a key QTL gene in rice that senses chilling, significantly enhanced the chilling tolerance of *japonica* rice with the overexpression of this gene, whereas functional deletion mutant COLD-1 and antisense gene lines were susceptible to chilling damage [6]. Zhang et al. found that three genes (LOC_Os01g55510, LOC_Os01g55350, and LOC_Os01g55560) in chromosome 1 of rice were closely associated with chilling tolerance [19], indicating the complexity of plant tolerance to chilling. In addition, several transcription factors have been identified, including *MYBS3*, *OsWRKY71*, and *OsWRKY76*, which play important roles in the tolerance of rice to chilling [20,21,22]. Recently, the role of calcium-dependent protein kinases in the chilling tolerance of rice has been widely reported [21,23,24]. These results provided important theoretical references and a scientific basis to further study and understand the molecular mechanisms and regulatory network of rice tolerance to chilling.

Improving the cold tolerance of rice seedlings under low-temperature conditions is very important in promoting rice growth. The use of biostimulants to counteract the effect of abiotic stress is well-recognized [25,26]. However, biostimulants are plant extracts and their bioactive compounds and tolerance mechanisms are unknown [27,28]. The most common components of the biostimulants are mineral elements, humic substances (HSs), vitamins, amino acids, chitin, chitosan, and poly- and oligosaccharides [29]. In addition, biological stimulants positively affect enzyme activity and gene expression in plants’ primary and secondary metabolism [29]. Therefore, biostimulins can improve plant tolerance to extreme temperature, which is an important means and hope for alleviating cold damage in sustainable agriculture [30,31].

Dular was a landrace *aus* variety that originated from India [32,33]. Dular exhibited significant chilling injury at temperatures below 15 °C [34,35]. The damage to plants under chilling is closely related to the intensity and time of stress. The tolerance of plants to external stress has a limit. The damage caused by short-term stress is reversible, and plants can adapt and return to normal growth. If the stress time is too long or the level of stress is too high, irreversible damage occurs, damaging or killing the plant. The cold spell in late spring is a common but unavoidable climate phenomenon in China. The sudden drop in temperature in a short period causes serious harm to crop growth. In real field production, such temperature changes are often experienced (normal temperature–chilling–restore). In this study, we analyzed rice plants’ physiological response and transcriptional regulation during three continuous and short periods of normal temperature–chilling–recovery to normal temperature. This study confirms the mechanism of hypothermic sensitivity reversal in Dular seedlings and provides knowledge on chilling tolerance mechanisms of rice.

## 2. Results

### 2.1. Changes in Chlorophyll and Proline and Non-Structure Carbohydrate (NSC) Contents of Rice Leaves

The chlorophyll and proline contents of the Dular were significantly different between normal-temperature treatment (CT), chilling treatment (TR), and recovery to normal-temperature treatment (RC) (Figure 1A). The chlorophyll a+b and carotenoid content of the Dular were significantly reduced under TR (0.28 mg·g^−1^ Fresh Weight (FW) and 0.01 mg·g^−1^ FW) as compared with CT (0.83 mg·g^−1^ FW and 0.07 mg·g^−1^ FW). On the one hand, the chlorophyll a+b and carotenoid content in RC (0.37 and 0.03 mg·g^−1^ FW) was significantly higher than that in TR, although it was lower than in CT (Appendix A). On the other hand, the proline content in RC (38.08 μg·g^−1^·FW) was significantly lower than that in TR (55.60 μg·g^−1^·FW), although higher than CT (29.13 μg·g^−1^·FW) (Figure 1B). The NSC, including the soluble sugar content (SSC) and starch content (SC) in RC (8.44 mg·g^−1^ Dry Weight (DW) and 35.69 mg·g^−1^ DW) were significantly higher than TR (4.65 mg·g^−1^ DW and 27.94 mg·g^−1^ DW), although lower than CT (17.86 mg·g^−1^ DW and 60.40 mg·g^−1^ DW). These findings indicated that damage occurred in the leaves under TR and recovered quickly under RC, though they differed from CT.

### 2.2. Changes in K^+^ and Na^+^ Flux Rate of Rice Roots

The root meristem region of rice was measured in this study. There were significant differences in the ion flow rate of rice roots under three different treatments. K^+^ efflux under TR (31.37 pmol·cm^−2^·s^−1^) was significantly higher than CT (6.11 pmol·cm^−2^·s^−1^) and RC (6.43 pmol·cm^−2^·s^−1^) (Figure 2A). There was no significant difference between RC and CT. The flux rate of Na^+^ revealed an opposite trend. Roots absorbed a small amount of Na^+^ under TR (−5.64 pmol·cm^−2^·s^−1^), whereas CT (24.34 pmol·cm^−2^·s^−1^) and RC (140.28 pmol·cm^−2^·s^−1^) excreted a large amount of Na^+^ externally (Figure 2B). The efflux rate of RC under RC was significantly higher than that of CT. These results indicated that the ion balance in Dular was affected by chilling stress, resulting in the efflux of K^+^ and the absorption of Na^+^, thereby disrupting the K^+^/Na^+^ balance. When the temperature was restored, plants could effectively inhibit K^+^ efflux and excrete Na^+^ in large quantities.

### 2.3. Gene Identification

A total of 21,970 genes were identified in all CT, TR, and RC samples (Figure 3). Among these, 20,096, 20,177, and 19,209 genes were identified in CT, TR, and RC samples, respectively (Appendix A). Furthermore, 839, 733 and 444 genes were identified in CT and TR, TR and RC, and CT and RC, respectively. A total of 17,748 genes were synchronously expressed across all three treatments (CT, TR, and RC).

### 2.4. Correlation between Different Temperature Treatments in Dular Samples

The principal component analysis (PCA) (Figure 4) of the three treatments depicted the high reproducibility in transcriptomic data. The difference between TR and CT samples was significant, indicating that the transcriptomic of Dular significantly changed under chilling, and the plants were subjected to intense stress. However, the differences between RC and CT samples were reduced than those between TR and CT samples, suggesting that the increase in temperature effectively mitigated the damage of chilling on Dular.

### 2.5. Transcriptomic Response to Chilling and Normal Temperature Recovery in the Leaves of Dular

A total of 11,732 genes were differentially expressed by expression ratio >2.000 or <0.050 during chilling (TR vs. CT (TR/CT)) and normal-temperature recovery (RC vs. TR (RC/TR)). Among those, 9143 genes were differentially expressed in TR/CT, and 6091 genes were differentially expressed in RC/TR. A total of 3502 genes were differentially expressed in both TR/CT and RC/TR (Appendix A). Furthermore, 5524 differentially expressed genes (DEGs) were upregulated and 3619 DEGs were downregulated in TR/CT. Moreover, 2789 DEGs were upregulated and 3302 were downregulated in RC/TR (Appendix A).

### 2.6. KEGG Pathway Enrichment under Chilling and Subsequent Recovery

KEGG pathway enrichment analysis was performed using an R script for genes in gene concentration. The KEGG pathway was significantly enriched when the *p* value was < 0.05. In addition, KEGG enrichment analysis was carried out for differentially expressed genes from normal temperature to chilling (TR/CT) (Figure 5A) and their subsequent recovery to normal temperature (RC/TR) (Figure 5B). The results demonstrated that ten metabolic pathways were significantly enriched under chilling and subsequent recovery, including alpha-Linolenic acid metabolism, carbon fixation in photosynthetic organisms, diterpenoid biosynthesis, glyoxylate, and dicarboxylate metabolism, linoleic acid metabolism, phenylpropanoid biosynthesis, photosynthesis-antenna proteins, ribosome, ribosome biogenesis in eukaryotes, and starch and sucrose metabolism (Figure 6).

### 2.7. DEGs Enriched for KEGG under Chilling and Subsequent Recovery

Enrichment analysis demonstrated that DEGs were mainly enriched in response to the ribosome pathway (map03010), photosynthesis-antenna proteins (map00196), phenylpropanoid biosynthesis (map00940) and linoleic acid metabolism (map00591). There were 103 DEGs (58 upregulated and 45 downregulated) and 144 (71 upregulated and 73 downregulated) in TR/CT and RC/TR (Appendix A). There were 5 DEGs (3 upregulated and 2 downregulated) and 46 DEGs (36 upregulated and 10 downregulated) responding to only TR/CT or RC/TR, respectively, whereas there were 97 DEGs in both TR/CT and RC/TR. Among these 97 DEGs, 11 DEGs were upregulated in TR/CT and RC/TR, whereas 19 DEGs were downregulated both in TR/CT and RC/TR. The heatmap illustrates the common genes (Figure 7A) and total genes (Appendix A) under three different temperature treatments. In addition, 44 DEGs were upregulated in RT/CT and downregulated in RC/TR, and 23 DEGs were downregulated in TR/CT and upregulated in RC/TR (Figure 7B). Moreover, 148 DEGs demonstrated 156 subcellular localization results, which indicated that some proteins might locate in multiple organelles. There were 46 (29.30%), 16 (10.19%), 16 (10.19%), 5 (3.18%), 11 (7.01%), 4 (2.55%), 16 (10.19%), 8 (5.73%) and 34 (21.66%) located in the chloroplast, cytoplasm, cytosol, endoplasmic reticulum, membrane, mitochondrial, nucleus, unknown and other regions, respectively (Figure 7C). These results indicated that many chloroplast genes are related to TR/CT and RC/TR.

### 2.8. DEGs Analysis of Common Responsive Genes between TR/CT and RC/TR

A total of 98 DEGs demonstrated a specific response in TR/CT and RC/TR. There were 44 genes upregulated in TR/CT and downregulated in RC/TR. Twelve of these genes were categorized as being related to phenylpropanoid biosynthesis, including seven peroxidase, two beta-glucosidase, cinnamoyl-CoA reductase 1, cytochrome P450, cinnamyl alcohol dehydrogenase 7. Five genes were categorized as being related to ribosomal, including 60S ribosomal protein L23, 60S ribosomal protein L32-1, syntaxin-61, 60S ribosomal protein L31, and 40S ribosomal protein S23. Moreover, genes related to linoleic acid metabolism included one phospholipase A2 homolog 3.

There were 24 genes downregulated in TR/CT, although they were upregulated in RC/TR. Genes related to linoleic acid metabolism included cytochrome P450 71D8, ent-isokaurene C2-hydroxylase, cytochrome P450 71A1, 3 lipoxygenase (2 lipoxygenases 2.3 and 1 lipoxygenase), linoleate 9S-lipoxygenase 6, and triacylglycerol lipase SDP1. Moreover, genes related to phenylpropanoid biosynthesis included one 3-aminomethylindole N-methyltransferase and one 4-coumarate-CoA ligase-like 2. One nucleolar protein 5-2 was associated with the ribosome.

Only 11 genes were upregulated in both TR/CT and RC/TR. Genes related to linoleic acid metabolism included 1 cytochrome P450 71D7. In addition, the transferase family protein was related to phenylpropanoid biosynthesis. In addition, 19 genes were downregulated in both TR/CT and RC/TR. Genes related to ribosomes included one 50S ribosomal protein L3-2 and one 50S ribosomal protein L25.

### 2.9. Analysis of Genes Only Responsive in TR/CT or RC/TR

Only five genes were differentially expressed in TR/CT; three were upregulated, and two were downregulated. Chlorophyll a–b binding protein CP24 upregulated was associated with photosynthesis—antenna proteins, and one ent-isokaurene C2-hydroxylase was downregulated due to linoleic acid metabolism.

In total, 36 genes which were only upregulated in RC/TR. Genes related to linoleic acid metabolism included 3 lipoxygenases and 1 ent-cassadiene hydroxylase. One beta-glucosidase 15 was regulated due to phenylpropanoid biosynthesis. Genes related to photosynthesis-antenna proteins included 10 chlorophyll a-b binding proteins.

Ten proteins were downregulated only in RC/TR. Genes related to the ribosome included 60S ribosomal protein L37-1, 60S ribosomal protein L9, RP-L32, ubiquitin-60S ribosomal protein L40-2, 40S ribosomal protein S10-2, and 40S ribosomal protein S13-2.

### 2.10. The Networking Analysis of Genes Responsive to Chilling and Subsequent Recovery

The Dular leaves were enhanced in response to exogenous chilling stimuli, as evidenced by the downregulation of the pathway of linoleic acid metabolism and the upregulation of the pathways of phenylpropanoid biosynthesis and ribosome. When plants recovered from chilling to normal temperature, leaves changed from tolerance to chilling to growth and development, the linoleic acid metabolism and photosynthesis pathway were upregulated, and the pathway of phenylpropanoid biosynthesis and ribosome were downregulated (Figure 8). qRT-PCR results also confirmed that the gene expression in linoleic acid metabolism and ribosome synthesis were consistent with the transcriptome results (Figure 9). The path of leaves’ responses to chilling and subsequent recovery is displayed in Figure 10.

## 3. Discussion

### 3.1. Leaves Responses to Temperature Change

Many plants suffer from physiological dysfunction after being subjected to chilling stress, adversely affecting their growth and development [36,37]. Although plants may acquire chilling tolerance (i.e., chilling adaptation) in their initial response to low but nonfreezing temperatures, continuous chilling exposure is considered catastrophic. Chloroplast plays an important role in rice development. Under chilling, rice leaves revealed symptoms of chilling injury and chlorophyll synthesis was blocked, leading to decreased photosynthesis [38,39,40]. Photosynthesis is an essential factor for rice yield and the most obvious physiological processes in rice under chilling [41]. Compared with chilling sensitive rice, chilling tolerant rice has excellent characteristics of tolerance and adaptation to cold damage caused by low temperature. According to our previous study wild-type Dular rice was sensitive to low temperatures when the ambient temperature was below 15 °C, and the leaves of this rice turned white in a short period [34,35]. Hence, Dular was used for all experiments in this study. This study found that the chlorophyll content of Dular leaves decreased significantly under chilling, whereas it recovered quickly during normal temperature, indicating that the Dular plants were vulnerable regarding chilling tolerance and susceptibility to environmental temperature. Plants produce many reactive oxygen species (ROS) under stress [42]. ROS are important signaling molecules in plant growth, development, and response to various stresses [43]. It has been reported that proline protects enzymes from denaturation and helps stabilize the machinery of protein synthesis, regulates the cytosolic acidity, enhances water-binding capacity, and acts as a reservoir of carbon and nitrogen sources [44]. Under stress, proline biosynthesis was activated in plants, and the proline degradation was inhibited, resulting in a large amount of proline accumulation in plants. Meanwhile, these proline could be decomposed and utilized quickly after the stress is relieved [45]. In this study, the proline content in TR was significantly higher than that in CT, which protected the rice system leaves, stabilized the cell membrane’s function, effectively eliminated ROS damage to the plant body, and improved the chilling tolerance of the plants. The proline content in RC was significantly higher than in TR, indicating a reduced chilling influence on the leaves. Large amounts of proline are converted to carbon and nitrogen sources. Meanwhile, the NSC content was reduced in the TR and rapid increase in RC. This result indicated that rice increased its tolerance under chilling stress by consuming a lot of NSC, and when the stress was relieved, rice recovered rapidly and synthesized a large amount of NSC for the body’s nutritional needs. 

Maintaining K^+^/Na^+^ balance is vital for plants to adapt to stressful environments [46,47]. Under stress, maintaining the balance of K^+^/Na^+^ in the cytoplasm mainly includes two aspects: avoiding the excessive accumulation of Na^+^ in the cytoplasm by promoting Na^+^ efflux; limiting the loss of K^+^ [48,49]. In this study, K^+^/Na^+^ in rice roots was balanced under CT, while under TR, a large amount of Na^+^ was absorbed internally, and K^+^ was excreted externally, resulting in disruption of intracellular ion balance and impaired cell membrane function. Under RC, plants effectively inhibit K^+^ efflux and excrete Na^+^ in large quantities, indicating that RC effectively maintains the intracellular ion balance and alleviate the damage caused by low temperature stress.

These results indicated that rice sensed and transmitted signals, stimulating its chilling tolerance function through physiological response. However, the chilling sensitivity of rice is a complex biological process, that involves changes at both physiological and transcriptional regulation levels. Gene expression drives a series of cellular activities and regulating of these requires the interaction of transcription factors (proteins) and transcription factor-binding sites (DNA elements). Transcriptional regulation is not a simple independent process, but a highly interactive gene regulatory network composed of hundreds of transcription factors, target sequences, and co-regulators.

### 3.2. DEGs Involved in the Pathway of Photosynthesis

This study found that the photosynthetic pathway in the Dular was significantly improved during the transition from chilling to normal-temperature recovery and was enriched in all metabolic pathways, indicating that the activation of the photosynthetic pathway played an important role in the removal of chilling sensitivity in Dular. Photosynthesis is the basis of life on earth. The leaves of higher plants capture light energy mainly through chlorophyll a/b-binding protein, and most of the chlorophyll is located in PSI and PSII natural pigment-protein complexes [50]. Light-trapping regulation is necessary for photosynthesis to maximize the use of light energy under low light exposure and avoid damage to photosynthetic apparatus under high light exposure [51,52]. Chlorophyll a/b-binding proteins play an important role in regulating plant light capture. Sun et al. found that chlorophyll a/b-binding proteins in winter wheat leaves were downregulated under chilling [53], thus affecting photosynthesis. Wang et al. found that under chilling, chlorophyll a/b-binding proteins could induce the upregulated expression of photosynthesis-related proteins, enzymes, and development-related proteins to enhance plant tolerance [54]. All chlorophyll a/b-binding proteins in Dular’s leaves were upregulated at normal temperature, and photosynthesis increased to produce more photosynthetic products for nutritional needs. The chlorophyll content also supported this idea. This result indicated that the temperature recovery could enhance photosynthesis and alleviate the effects of chilling, further confirming the transcriptomic results.

### 3.3. DEGs Involved in the Pathway of Linoleic Acid Metabolism

Linoleic acid acts as a structural component to maintain the transdermal water barrier of the epidermis membrane fluidity. In addition, when released from membrane phospholipids, it can be enzymatically oxidized to a various derivatives involved in cell signaling [55]. Cytochrome P450 is a metabolic enzyme widely existing in aerobic organisms, which catalyzes and regulates the metabolism and transformation of endogenous substances or the activation and degradation of exogenous compounds in life [56,57]. Lipoxygenases catalyze the oxidation of polyunsaturated fatty acids to produce unsaturated fatty acid and hydroperoxides; this process is known as the hydroperoxidation of lipids [58]. Lipids, primarily stored as triacylglycerols (TAGs) in seed, are the most energy-dense and a common form of seed reserves to support seedling establishment [59]. TAG lipases first hydrolyze TAG stored in seeds to release free fatty acids (FAs) that are transported into peroxisomes and oxidized to acetyl-CoA via multi-ple rounds of β-oxidation cycles [60]. In this study, the expression levels of cytochrome (P450 71D8 and P450 71A1), lipoxygenase, linoleate 9S-lipoxygenase 6 and triacylglycerol lipase SDP1 were downregulated in TR/CT and upregulated in RC/TR, indicating that the plant was highly stressed. The material and energy circulation were obstructed under chilling, whereas the stress was relieved when the temperature subsequent reverted to normal. The linoleic acid metabolism upregulated in RC/TR is beneficial for plants to alleviate the damage caused by chilling.

### 3.4. DEGs Involved in the Pathway of Phenylpropanoid Biosynthesis

Phenylpropanoid biosynthesis plays a vital role in plant stress tolerance [61], including phenylpropanoid, coumarin, lignin, and flavonoids. Peroxidase is a redox enzyme produced by plants, and it can catalyze many reactions and has the dual function of eliminating the toxicity of hydrogen peroxide and phenols, amines, aldehydes, and benzene [62]. Peroxidase is widely found in plants and is involved in hormone metabolism [63], reactive oxygen signaling molecule metabolism [64], and plant tolerance [65]. β-glucosidase is an important hydrolytic enzyme [66], and it is involved in glucose metabolism and plays an important role in maintaining the normal physiological function of an organism. One study also found that β-glucosidase had biological significance in plants’ virulence, defense, and secondary metabolisms [67]. This study found that phenylpropyl biosynthesis was significantly enhanced in Dular under chilling, in which β-glucosidase and peroxidase were in the highest proportion, and the expression was upregulated overall. The results demonstrated that the tolerance of Dular to chilling was enhanced. The phenylpropane biosynthesis pathway in the Dular was downregulated during the transition from chilling to normal-temperature recovery. This was the exact opposite during the transition from normal-temperature recovery to chilling treatment in Dular. The results revealed that the phenylpropyl biosynthesis pathway was upregulated in Dular in the transition from normal temperature to chilling due to the effect of chilling on the stress tolerance of rice plants. However, when plants recovered from chilling to normal temperature, the phenylpropanoid biosynthesis pathway in the organism was downregulated, and the plant activity changed from resisting chilling to growth and development.

### 3.5. DEGs Involved in the Pathway of Ribosome

Ribosome synthesis is a key to plant growth and environmental adaptation [68]. For example, Moin et al. found that the ribosomal protein L6 enhanced the salt tolerance of rice [69]. Ribosomes are the oldest and most sophisticated organelles, and their structure and composition remain highly conserved from prokaryotic to eukaryotic. Ribosomes are a vital part of cell replication, responsible for protein synthesis, and play an important role in controlling cell growth, division, and development [70]. A large number of studies have demonstrated that in addition to forming ribosomes and participating in protein biosynthesis [71], ribosomal proteins are involved in chloroplast development [72], translation regulation [73], and other processes. This study further analyzed the downregulated ribosome synthesis pathway with significant enrichment. Ribosome biogenesis involves the transcription of ribosomal DNA (rDNA), precursor-rRNA (pre-rRNA) processing, RNA modifications, and assembly factors [74,75]. The ribosome synthesis in Dular plants under chilling was upregulated, enhancing the synthesis of other proteins and reducing rice damage.

## 4. Materials and Methods

### 4.1. Plant Growth and Treatment

In our previous study, wild-type Dular rice was sensitive to low temperatures when the ambient temperature was below 15 °C, and the leaves of this rice turned white in a short period of time [34,35]. Hence, Dular was used for all experiments in this study. 

The plump rice seeds were selected, disinfected with 25% sodium hypochlorite solution for 30 min, washed with deionized water, and soaked for 24 h. Then, the seeds were put into an incubator for 24h. After germination, the seeds were evenly planted on the trays filled with matric soil (PINDSTRUP, http://www.pindstrup.cn/, accessed on 15 August 2022), and the trays were placed in the incubator (Yiheng, Shanghai, China). The culture conditions were as follows: 28 °C/day for 14 h, the light intensity of 20,000 Lux, dark culture at 22 °C/night for 10 h, and humidity of 85%. When the rice grew to the stage with 1.5 leaves, three continuous and short periods of treatments were carried out: 

Normal-temperature treatment (CT): The culture conditions were 28 °C/day for 14 h, light intensity of 20,000 Lux, and 22 °C/night for 10 h. The treatment period was 24 h.

Chilling treatment (TR): Rice plants of CT were transferred to 15 °C/day for 14 h, with a light intensity of 20,000 Lux, and 10 °C/night for 10 h. The treatment period was 24 h.

Return to normal-temperature treatment (RC): Rice plants of TR were transferred to 28 °C/day for 14 h, with a light intensity of 20,000 Lux, and 22 °C/night for 10 h. The treatment period was 24 h.

The treatment sequence is as follows: normal temperature treatment for 24 h (sampling), then transferred to chilling treatment for 24 h (sampling), and finally transferred to normal temperature treatment for 24 h (sampling). The entire treatment took three days.

### 4.2. Determination of Chlorophyll, Proline and NSC Content

#### 4.2.1. Determination of Chlorophyll Content

The fresh leaves (0.2 g, three biological repetitions) were grounded with 96% ethanol to a final volume of 25 mL, as proposed by Sadia et al. [35] and Wang [76]. The absorbance value of the extract at the wavelengths of 665 nm, 649 nm and 470 nm was determined using an Infinite M200 PRO NanoQuant (Tecan, Männedorf, Switzerland) microplate reader. As a blank control, 96% ethanol was used. Finally, the content of chlorophyll in plant tissues was calculated according to the following formula: Chlorophyll content (mg/g) = [concentration of chlorophyll × volume of extraction liquid × dilution ratio]/dry weight of the sample.

#### 4.2.2. Determination of Proline Content

The content of the proline was measured using the kit according to the product manual (Cominbio, www.cominbio.com, accessed on 15 August 2022). The contents of proline were expressed as μg·g^−1^·FW. The determination principle was as follows: proline was extracted with sulfosalicylic acid. After heat treatment, proline reacted with acidic ninhydrin solution to produce a red color. After adding toluene, the absorbance was measured at 520 nm and finally converted to proline content.

#### 4.2.3. Determination of NSC Content

The content of NSC was measured by Wang’s method [76] as follows: the leaves were dried, crushed, and subjected to a 100-mesh sieve. Samples were weighted (0.2 g, three biological repetitions) and placed in glass test tubes. Next, 5 mL of deionized water was added to the test tubes and heated in a 100 °C boiling water bath for 30 min. The solution was filtered into a 50 mL volumetric flask, and the volume was fixed. Then, 0.5 mL extract was transferred from a 50 mL volumetric flask to a 20mL glass test tube, then added to 5mL anthrone-sulfuric acid solution (0.2% *w*/*v* anthrone dissolved in sulfuric acid) and heated in 100 °C boiling water for 15 min. After cooling, the absorbance at 630 nm was measured and finally converted into soluble sugar content.

After the soluble sugar extraction, the residue was transferred to a new 50 mL glass test tube. Then, 10 mL of deionized water was added and heated in the 100 °C boiling water for 15 min. Then 5 mL 9.2 M perchloric acid solution was added to a boiling water bath and heated for 15 min. Finally, the volume was fixed to 50 mL. Next, 0.5 mL extract was transferred from a 50 mL volumetric flask to a 20 mL glass test tube, then 5 mL anthrone-sulfuric acid solution (0.2% *w*/*v* anthrone dissolved in sulfuric acid) was added and heated in a 100 °C boiling water bath for 15 min. After cooling, the absorbance at 630 nm was measured and finally converted into starch content.

### 4.3. The Measurement of K^+^ and Na^+^ Fluxes

The non-invasive Micro-test Technology (NMT-YG-100, Younger USA LLC, Amherst, MA, USA) and V2.0 Software (Younger USA LLC, Amherst, MA, USA) was used to measure the flux rates of K^+^ and Na^+^ [77]. The technique was based on the nondestructive microelectrode technology, through the automatic control of a computer and precise motion control system, without touching the sample, the three-dimensional, real-time, and dynamic measurement, the sample in and out of various ion flow rates and their motion direction [78]. The roots were immobilized in a culture dish (diameter: 6 cm) and calibrated with a measuring solution for 10 min. The microelectrode (Younger USA LLC, Amherst, MA, USA) was vibrating between the root surface and measuring solution (two positions, i.e., 30 µm) and kept a perpendicular axis to the root (Figure 2A). The commercially available ionophore (K^+^ and Na^+^) (Younger USA LLC, Amherst, MA, USA) was filled in the microelectrode tip. The measuring solution for K^+^ consisted of 0.1mM CaCl_2_, 0.1mM KCl and 0.3mM 2-(N-Morpholino) ethanesulfonic acid (MES), pH 6.0. The measuring solutions for Na^+^ consisted of 0.1mM CaCl_2_, 0.1mM KCl, 0.1mM NaCl and 0.3mM MES, pH 6.0. The positive value of flow velocity data refers to efflux, while the negative value refers to the influx.

### 4.4. RNA Extraction

Total RNA of Dular-CT, Dular-TR and Dular-RC was extracted using TRIzol (TransGen Biotech, Beijing, China). The concentration of RNA was determined by Nanodrop 2000 C (Thermo Fisher Scientific, Waltham, MA, USA). RNA was stored in the Ultra-low temperature Freezer DW-HL678 (Meiling, HeFei, China) at −80 °C.

### 4.5. Transcriptome Sequencing

Eukaryotic mRNA sequencing was based on Illumina Novaseq 6000 sequencing platform (Meiji, http://www.majorbio.com/, accessed on 15 August 2022), and all mRNA of samples were sequenced. An Illumina Truseq RNA Sample Prep Kit (Meiji, http://www.majorbio.com/, accessed on 15 August 2022) was used to construct the gene pool. The operation flow and instrument reagents were as follows: RNA integrity was detected by agarose gel electrophoresis. The gene pool was constructed based on ensuring the RNA quality of the samples (total content of RNA ≥ 1 μg, content ≥ 35 ng/μL, OD260/280 ≥ 1.8, OD260/230 ≥ 1.0). First, the poly(A) tail structure at the 3’ end of a eukaryotic mRNA was exploited, and the poly(A) mRNA with poly(A) tail was isolated by A-T base pairing using magnetic beads with Oligo (dT) (Meiji, http://www.majorbio.com/, accessed on 15 August 2022). The Illumina Novaseq 6000 platform was designed to sequence short sequences. The enriched mRNA is a complete RNA sequence with an average length of several kb, which requires random interruption. Then mRNA was fragmented by adding fragmentation buffer, and small fragments of 300 bp were screened through magnetic beads to reverse cDNA synthesis. Under reverse transcriptase, random hexamers were added to synthesize one-strand cDNA using mRNA as a template, and then two-strand synthesis was carried out to form a stable double-strand structure. The cDNA structure of the double-strand was the vicious end. In addition, End Repair Mix was added to make the flat end, and then an “A” base was added to the 3’ end to connect the Y-shaped connector. The target strip was then enriched and recovered by the library. After the TBS380 quantitative analysis, the corresponding data were mixed and sequenced on the machine.

### 4.6. Bioinformatics Analysis of Transcriptomic Data from Oryza sativa L.

Quality control was carried out on the original sequencing data using fastp (http://github.com/OpenGene/fastp, accessed on 15 August 2022) before analysis to obtain high-quality clean data to ensure the accuracy of subsequent analysis results. The genome of *Oryza sativa* L. ssp. Indica was set as a reference. The mapped data for subsequent transcript assembly and expression volume calculations were obtained using TopHat2 software (http://ccb.jhu.edu/software/tophat/index.shtml, accessed on 15 August 2022). Using StringTie software (http://ccb.jhu.edu/software/stringtie/, accessed on 15 August 2022) to concatenate mapped reads. NCBI_NR database (http://www.ncbi.nlm.nih.gov, accessed on 15 August 2022) and Swiss-PROt database (http://web.expasy.org/docs/swiss-prot_guideline.html, accessed on 15 August 2022) were used for functional annotation and statistics of transcripts. The unigene expression was calculated and normalized to Fragments Per Kilobase of transcript per Million mapped reads (FPKM) by using RSEM software (http://deweylab.github.io/RSEM/, accessed on 15 August 2022). Then Venn and Principal Component Analysis (PCA) were used for sample relationship analysis.

The edgeR software (http://bioconductor.org/packages/stats/bioc/edgeR/, accessed on 15 August 2022) was used for differential gene identification, with the default threshold of *p*-adjusted < 0.05 and |log2FC| ≥ 1. KEGG pathway (https://www.kegg.jp/, accessed on 15 August 2022) analysis of differential genes was performed using R software (The University of Auckland, Oakland, New Zealand). A *p* value of 0.05 was used as the threshold for the KEGG pathway enrichment analysis, and the metabolic pathways of the differentially expressed genes were determined.

MajorBio online platform (https://cloud.majorbio.com/page/tools/, accessed on 15 August 2022) was used to analyze the relationship between the different temperature treatments for Dular gene samples.

### 4.7. qRT-PCR Validation

RNA was reversely transcribed into cDNA using the cDNA Synthesis SuperMix (TransGen Biotech, Beijing, China). The CDS sequences of the target genes were retrieved from the Rice Genome Annotation Project Database (http://rice.uga.edu/index.shtml, accessed on 15 August 2022), and the upstream and downstream primers were designed based on the qRT-PCR primer database (https://biodb.swu.edu.cn/qprimerdb/blast, accessed on 15 August 2022). The qRT-PCR was carried out using the TransScript Green qRT-PCR Supermix kit (TransGen Biotech, Beijing, China) and the Eppendorf Realplex4 (Eppendorf, HAM, DE) instrument was used. The qRT-PCR conditions were as follows: predenaturation at 94 °C for 30 s, denaturation at 94 °C for 5 s, and extension at 60 °C for 30 s (45 cycles). Finally, the relative expression of each candidate mRNA in different samples was calculated by the 2^−ΔΔCt^ method. The qRT-PCR primers are listed in Appendix A.

### 4.8. Statistical Analysis

All of the data in the paper are the result of three biological replicates. The differences among the treatments were calculated and statistically analyzed using a one-way analysis of variance (ANOVA) by the least-significant-difference multiple-range test (LSD, *p* < 0.05). The statistical package for OriginPro 8.0 (OriginLab, Northampton, Ma, USA), GraphPad Prism version 5.0 (GraphPad, San Diego, CA, USA), and the Data Processing System (DPS) version 7.05 (Zhejiang University, Hangzhou, China) was used for the statistical analysis.

## 5. Conclusions

Low temperature severely affects the growth and development of rice. The recovery of temperature after a short period of low temperature ensured the synthesis of chlorophyll and NSC in rice and maintained the ion balance of the plant. The upregulation of photosynthesis, linoleic acid metabolism, and the downregulation of ribosome and phenylpropanoid biosynthesis indicate that plants changed from tolerance to stress to growth and development after temperature recovery. These results suggest that restoring normal temperature treatment can effectively reverse injury caused by chilling. This study revealed the process and mechanism of rice sensing, transmitting chilling signals, and stimulating the function of chilling tolerance, providing a theoretical foundation for further understanding the mechanism of chilling tolerance in plants and accelerating the progress of molecular breeding of chilling tolerant crops.

## Figures and Tables

**Figure 1 ijms-23-10739-f001:**
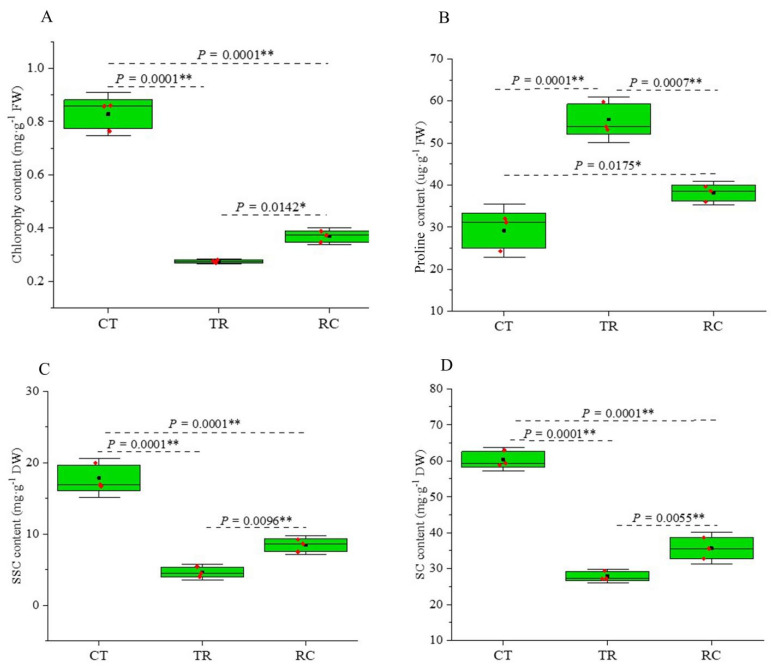
(**A**) The content of chlorophyll a + b under three different treatments. (**B**) The content of proline under three different treatments. (**C**) The SSC under three different treatments. (**D**) The SC under three different treatments. Error bars are the standard error (±SE) of three biological replications. * *p* < 0.05; ** *p* < 0.01 by Student’s *t*-test. CT: Normal-temperature treatment. TR: Chilling treatment. RC: Recovery to normal-temperature treatment. FW: Fresh weight. DW: Dry weight. SSC: Soluble sugar content. SC: Starch content.

**Figure 2 ijms-23-10739-f002:**
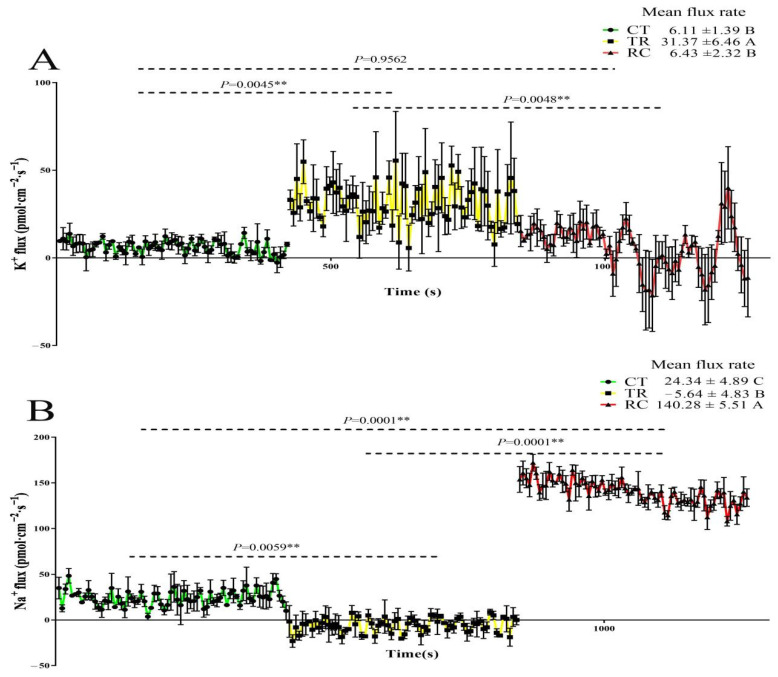
(**A**) The K^+^ flux rate of root under CT, TR and RC. (**B**) The Na^+^ flux rate of root under CT, TR and RC. Columns with different letters are significantly different (LSD test, *p* < 0.01). Error bars are the standard error (±SE) of three biological replications. CT: Normal-temperature treatment. TR: Chilling treatment. RC: Recovery to normal-temperature treatment. Positive flux rate indicates efflux. Negative flux rate indicates influx.

**Figure 3 ijms-23-10739-f003:**
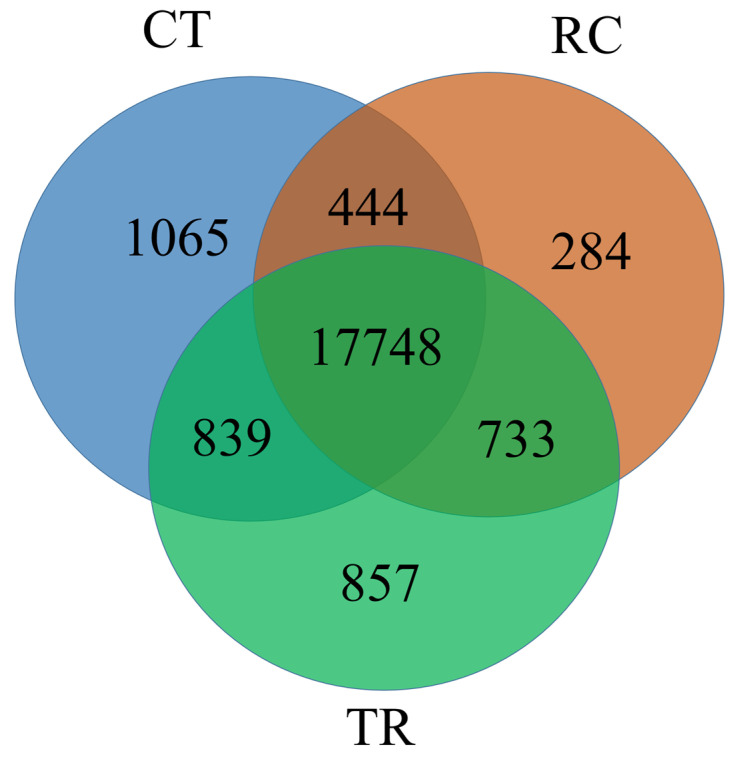
The Venn diagram between three treatments (CT, TR and RC). CT: Normal-temperature treatment. TR: Low-temperature treatment. RC: Recovery to normal-temperature treatment. The numbers in the figure represent the number of genes identified.

**Figure 4 ijms-23-10739-f004:**
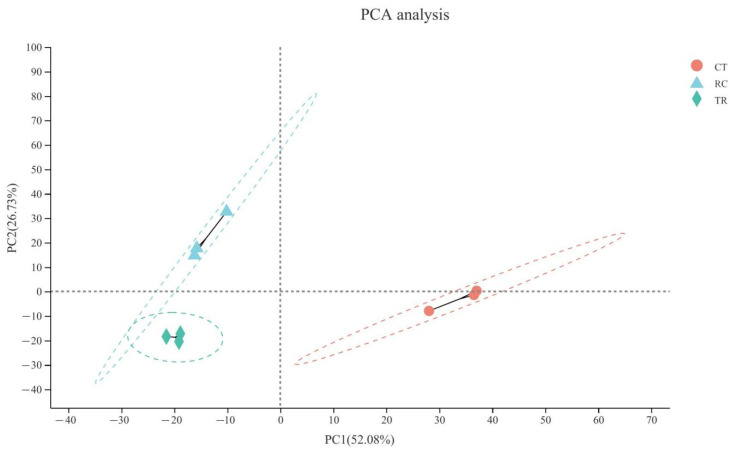
The principal component analysis (PCA) analysis between three samples. After dimensionality reduction analysis, the samples have relative coordinate points on the principal components. The distance of each sample point represents the sample distance, and the closer the distance is, the higher the similarity between samples. The horizontal axis represents the contribution degree of principal component 1 (PC1) in the two-dimensional graph to the distinguished samples, and the vertical axis represents the contribution degree of principal component 2 (PC2) in the two-dimensional graph to the distinguished samples. CT: Normal-temperature treatment. TR: Low-temperature treatment. RC: Recovery to normal-temperature treatment.

**Figure 5 ijms-23-10739-f005:**
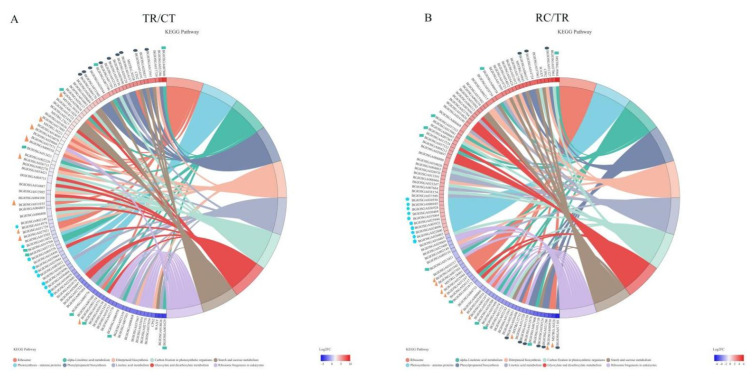
(**A**) The pathways were significantly enriched by KEGG analysis from normal temperature to chilling. (**B**) The pathways were significantly enriched by KEGG analysis from chilling recovery to normal temperature. The triangles, ellipses, pentagons and rectangles demonstrated the top enriched genes for KEGG in the ribosome, phenylpropanoid biosynthesis, photosynthesis-antenna proteins and linoleic acid metabolism, respectively.

**Figure 6 ijms-23-10739-f006:**
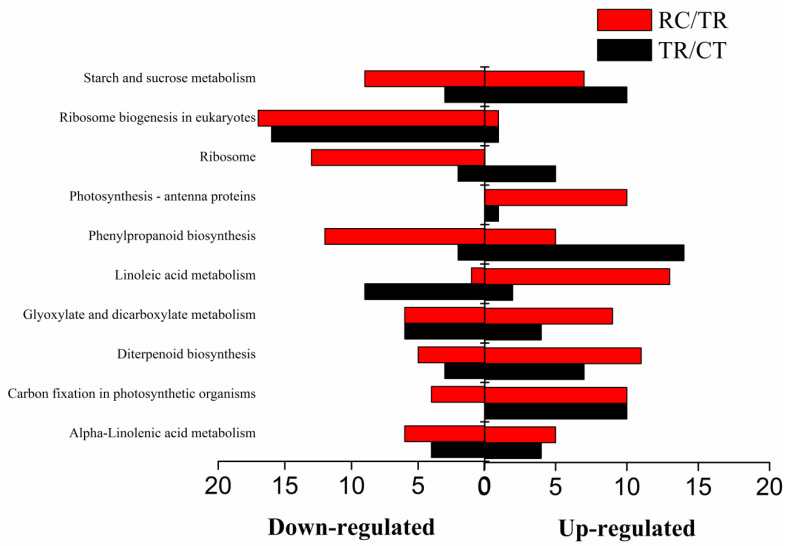
Functional characterization of chilling and recovery-responsive genes under chilling and/or subsequent recovery. CT: Normal-temperature treatment. TR: Chilling treatment. RC: Recovery to normal-temperature treatment. TR/CT: From normal temperature to chilling. RC/TR: From chilling recovery to normal temperature. The red columns show the differential expressed genes under RC/TR. The black columns show differential expressed genes under TR/CT.

**Figure 7 ijms-23-10739-f007:**
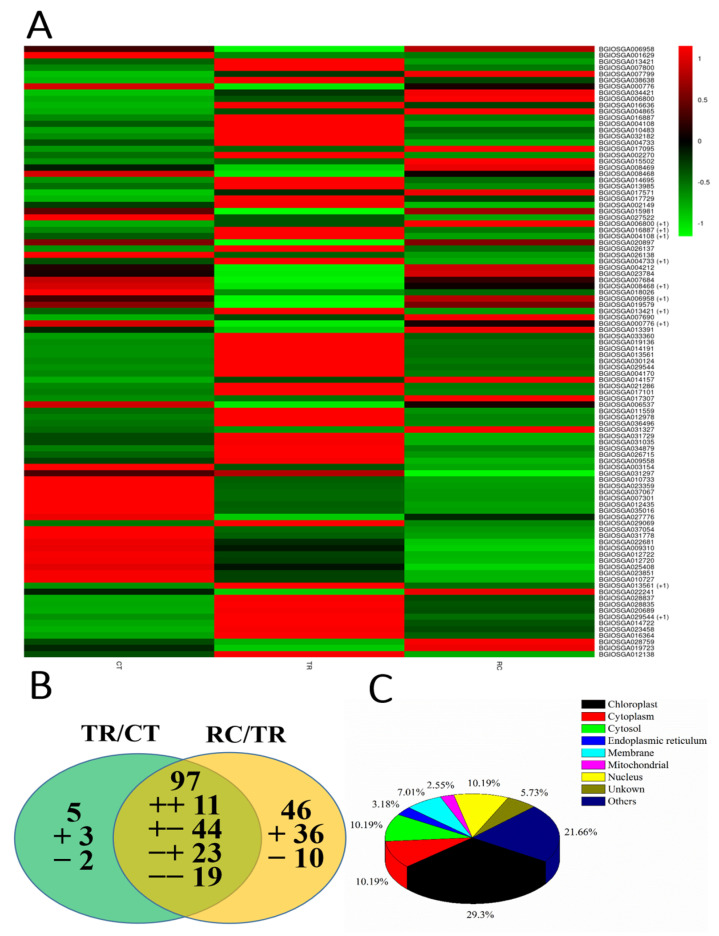
(**A**) Heatmap analysis of the common genes under three different temperature treatments. CT: Normal-temperature treatment. TR: Chilling treatment. RC: Recovery to normal-temperature treatment. (**B**) Venn diagram of differentially expressed genes that were upregulated or downregulated by chilling stress or recovery. TR/CT: From normal temperature to chilling. RC/TR: From chilling recovery to normal temperature. The green circle on the left shows that the genes are differentially expressed only in TR/CT. The intersection in the middle represents the genes are differentially expressed in both TR/CT and RC/TR. The yellow portion on the right shows the genes that are differentially expressed only in RC/TR. The “+” and “−” indicate upregulated and downregulated genes, respectively. The “+−” indicates genes with upregulated expression under TR/CT and RC/TR treatments. The “+−” indicates genes with upregulated expression under TR/CT and downregulated expression under RC/TR. The “−+” indicates genes with downregulated expression under TR/CT and upregulated expression under RC/TR. The “−−” indicates genes with downregulated expression under TR/CT and RC/TR. (**C**) Subcellular localization of 149 differentially expressed genes under chilling and/or subsequent recovery.

**Figure 8 ijms-23-10739-f008:**
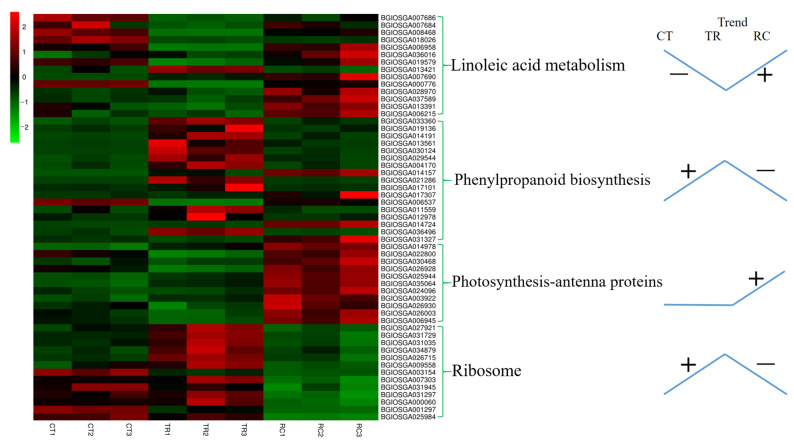
Heatmap analysis of the genes under three different temperature treatments. CT: Normal-temperature treatment. TR: Chilling treatment. RC: Recovery to normal-temperature treatment. CT1, CT2 and CT3 are the three biological replications of CT. TR1, TR2 and TR3 are the three biological replications of TR. RC1, RC2 and RC3 are the three biological replications of RC. The “+” and “−” indicate upregulated and downregulated pathway by chilling stress (TR/CT) or recovery (RC/TR), respectively.

**Figure 9 ijms-23-10739-f009:**
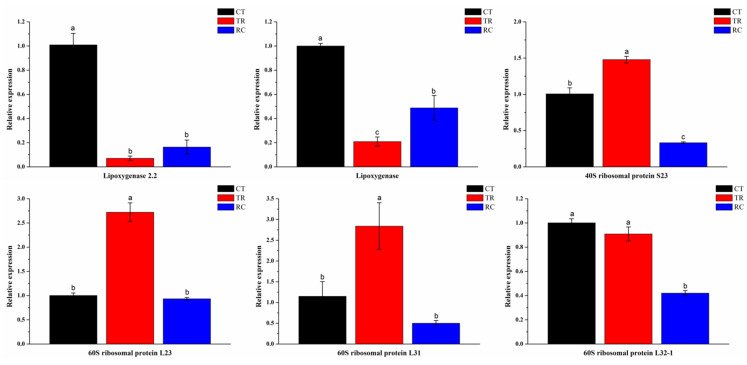
The qRT-PCR analysis shows the genes expression of Lipoxygenase 2.2, Lipoxygenase, 40S ribosomal protein S23, 60S ribosomal protein L23, 60S ribosomal protein L31 and 60S ribosomal protein L32-1 under three different temperature treatments. Columns with different letters are significant difference (LSD test, *p* < 0.01). Error bars are the standard error (±SE) of three biological replications. CT: Normal-temperature treatment. TR: Chilling treatment. RC: Recovery to normal-temperature treatment.

**Figure 10 ijms-23-10739-f010:**
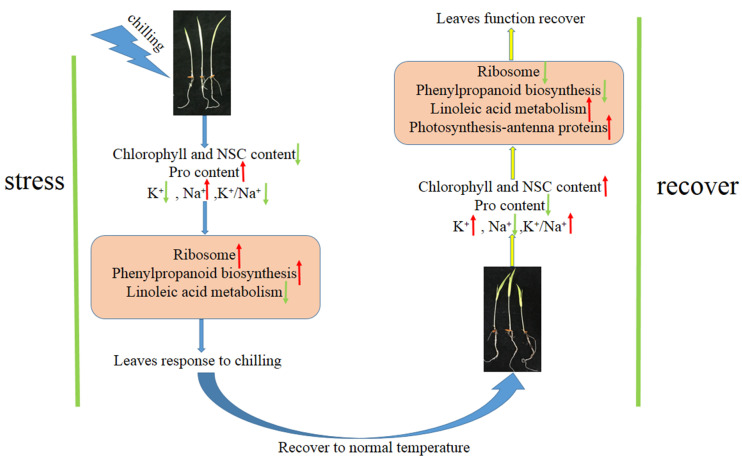
The pathway of leaves response to environmental conditions. Red arrows represent upregulation; green arrow represent downregulation.

## Data Availability

All data generated during this study are included in this published article and its Appendix A, and the raw data used or analyzed during the current study are available from the corresponding author on reasonable request.

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
