# Peer review of "Transcriptome Analysis of the Responses of Rice Leaves to Chilling and Subsequent Recovery"

_ijms, 2022, doi:10.3390/ijms231810739_

Round 1
Reviewer 1 Report (Previous Reviewer 1)
The article looks promising. I have few comments specially in relation to the figure presented.
1 Please treat every figure as an independent entity. I should not look back and forth to the text to understand the description of figure. Please include legends describing figure 7 b, figure 8 and figure 9 .
Author Response
Thanks for your suggestion. We have modified these in the revised manuscript.
Reviewer 2 Report (Previous Reviewer 3)
The manuscript entitled " Transcriptome analysis of the responses of rice leaves to chilling and subsequent recovery" and authored by Zhang Li and colleguagues, is the edited version of article I revised earlier. The authors correctly followed the suggestions, and now I believe the quality is greatly improved. Accordingly, I consider the manuscript to be publishable in IJMS.
Author Response
Thanks very much for your sincere comments and suggestions. Due to your constructive comments and suggestions, the study can be well developed.
This manuscript is a resubmission of an earlier submission. The following is a list of the peer review reports and author responses from that submission.
Round 1
Reviewer 1 Report
The manuscripts is well written. I found it well organized and I am satisfied with the presentation of the results and supporting discussion.
I have few suggestions mentioned below:
1- I would recommend removing Figure 2A from the text.
2- In figure 4, A and B are almost repetitive. If possible remove one.
Author Response
1: I would recommend removing Figure 2A from the text.
Answer:Thanks for your suggestion, we have removed the Figure 2A from the text.
2: In figure 4, A and B are almost repetitive. If possible remove one.
Answer:Thank you for your suggestion, we have removed the Figure 4A.
Reviewer 2 Report
Reviewer’s comment to the editor and Author
International Journal of Molecular Sciences
Dear editor and author,
I have thoroughly reviewed the manuscript titled “Transcriptome analysis of the responses of rice leaves to chilling and subsequent recovery”.
The present study was focused on investigating the transcriptome analysis of rice leaves in response to chilling stress and subsequent recovery.
The topic looks very interesting and timely topic but the presentation of the article looks choppy.
I recommend this article “major” and resubmit it again for further consideration.
The comments are mentioned below:
Overall comments:
§ Data presentation of some parts of the seems poor.
§ The findings are excellent but author failed to interpret it with logical and more organized manner.
§ Validation of the DEGs is not clear in the results section. I think qRT-PCR is required to validate the genes.
Abstract:
§ The abstract looks very poor.
§ Author didn’t present any result about the total genes and DEGs which is the very basic information of this study.
§ Line 29-31; based on what properties, author proposed that statement. Author should clarify it.
Introduction:
§ The font size throughout the manuscript looks choppy. Author should check it before further submission.
§ In line 66-68, author mention that many transcriptions factors may play crucial role towards chilling stress; but what is the relationship of this statement regarding the current topic because author didn’t highlight any transcription factor in this study.
Materials and Methods:
§ In RNA extraction and RNA-Seq section, author should write it more organized way. It is not clear at all. Some important information and procedures related to RNA sequencing; author should address in this section.
§ Author should provide detailed information about the accession line “Dular”.
§ Author may separate the Bioinformatics Analysis section. Author should provide details information of the databases that used to analyze the data.
Results:
§ Line 98; the findings value mentioned in the text regarding the figure looks different (CT (8.43 mg·g-1·DW and 35.69 mg·g-1·DW).
§ Line126; author should rewrite this deadline; it doesn’t make any sense.
§ Line 126-129; author should mention the basic information about the identified genes and their expression here.
§ Line 130-135; author should rewrite this paragraph. It is not clear at all. How many genes share each other is missing here.
§ Line 144-152; author must have to provide the supplementary tables regarding the information. Only figure is not enough to understand it easily.
§ Line 153-182; author should revise this section. Author should write it more organized way. Author may separate this section based on “GO Analysis”, “KEGG Pathway Analysis”, “Networking Analysis” etc.
§ Regarding GO and KEEG analysis, author may highlight the top GO-enriched genes.
§ Figure 3, author may highlight the top GO-enriched genes for KEGG.
§ Figure 5, author may highlight only common genes (Average data not replicate data). Total identified genes can be provide as a supplementary file.
Discussion:
§ The discussion part overall looks okay.
Conclusion:
§ Author should make a concluding remarks.
Data availability:
§ Basic information (Data) about the transcripts is missing here. Author should provide it as a supplementary file.
§ Author should add the up- and down-regulated genes information in a separate excel file.
§ Functional annotation with their enrichment analysis should be provided in a separate excel file.
§ KEGG pathway with their enrichment analysis should be provided in a separate excel file.
§ Author should provide a basic data file that contains genes information in a separate excel file.
However, I have addressed the major concerns about the article in the above. If author failed to address that issues, it is not possible to publish in this high esteemed Journal.
Thanks.
Author Response
- Data presentation of some parts of the seems poor.The findings are excellent but author failed to interpret it with logical and more organized manner.
Answer:Thank you for your suggestion, we have reorganized the data to present it in a better form. The logic and organization of this article are clearer than before.
- Validation of the DEGs is not clear in the results section. I think qRT-PCR is required to validate the genes.
Answer:Thank you for your suggestion, we have reorganized the data to present it in a better form. Some genes have been verified by qRT-PCR.
- The abstract looks very poor.
Answer:Thank you for your suggestion, we have reworked the abstract.
- Author didn’t present any result about the total genes and DEGs which is the very basic information of this study.
Answer:Thank you for your suggestion, we have added the result about the total genes and DEGs in the abstract. Transcriptome results showed that a total of 21,970 genes were identified at three different temperatures and 11,732 genes were different expressed.
- Line 29-31; based on what properties, author proposed that statement. Author should clarify it.
Answer:Thank you for your suggestion, KEGG enrichment analysis was carried out for DEGs in TR/CT and RC/TR. The results showed that 10 metabolic pathways were significantly enriched under chilling and subsequent recovery, including: alpha-Linolenic acid metabolism, carbon fixation in photosynthetic organisms, diterpenoid biosynthesis, glyoxylate and dicarboxylate metabolism, linoleic acid metabolism, phenylpropanoid biosynthesis, photosynthesis-antenna proteins, ribosome, ribosome biogenesis in eukaryotes and starch and sucrose metabolism. The pathway of photosynthesis-antenna proteins, ribosome, phenylpropanoid biosynthesis and linoleic acid metabolism were significantly difference between TR/CT and RC/TR. Base on this, we proposed this viewpoint.
- The font size throughout the manuscript looks choppy. Author should check it before further submission.
Answer:Thank you for your suggestion, we have unified the font size for the whole manuscript.
- In line 66-68, author mention that many transcriptions factors may play crucial role towards chilling stress; but what is the relationship of this statement regarding the current topic because author didn’t highlight any transcription factor in this study.
Answer:Thank you for your suggestion. Gene expression drives a series of cellular activities, and the regulation of these requires the interaction of transcription factors (proteins) and transcription factor-binding sites (DNA elements). Transcriptional regulation is not a simple independent process, but a highly interactive gene regulatory network composed of hundreds of transcription factors, target sequences, and co-regulators.
- In RNA extraction and RNA-Seq section, author should write it more organized way. It is not clear at all. Some important information and procedures related to RNA sequencing; author should address in this section.
Answer:Thank you for your suggestion, we have rewritten this section.
- Author should provide detailed information about the accession line “Dular”.
Answer:Thank you for your suggestion, we have provided information about the accession line “Dular”. Dular is a conventional indica (Oryza sativa L. ssp. Indica) rice introduced from India.
- Author may separate the Bioinformatics Analysis section. Author should provide details information of the databases that used to analyze the data.
Answer:Thank you for your suggestion, we have separated the Bioinformatics Analysis section and provided details information of the databases in “4.6. Bioinformatics analysis of transcriptomic data from Oryza sativa L.” that used to analyze the data. The details information of the databases that used to analyze the data as follow: TopHat2 software (http://ccb.jhu.edu/software/tophat/index.shtml); StringTie software (http://ccb.jhu.edu/software/stringtie/); NCBI_NR database (http://www.ncbi.nlm.nih.gov); Swiss-PROt database (http://web.expasy.org/docs/swiss-prot_guideline.html); RSEM software (http://deweylab.github.io/RSEM/); edgeR software (http://bioconductor.org/packages/stats/bioc/edgeR/); KEGG pathway (https://www.kegg.jp/).
- Line 98; the findings value mentioned in the text regarding the figure looks different (CT (8.43 mg·g-1·DW and 35.69 mg·g-1·DW).
Answer:Thank you for your suggestion. This is our data input error, we have corrected it.
- Line126; author should rewrite this deadline; it doesn’t make any sense.
Answer:Thank you for your suggestion, we have rewrote the deadline.
- Line 126-129; author should mention the basic information about the identified genes and their expression here.
Answer:Thank you for your suggestion, we have mentioned the identified genes and their expression.
- Line 130-135; author should rewrite this paragraph. It is not clear at all. How many genes share each other is missing here.
Answer:Thank you for your suggestion, we have rewrote this paragraph, and we have provided the supplementary tables S2 regarding the information.
- Line 144-152; author must have to provide the supplementary tables regarding the information. Only figure is not enough to understand it easily.
Answer:Thank you for your suggestion, we have provided the supplementary tables S3 regarding the information.
- Line 153-182; author should revise this section. Author should write it more organized way. Author may separate this sectionbased on “GO Analysis”, “KEGG Pathway Analysis”, “Networking Analysis” etc.
Answer:Thank you for your suggestion, we have separated this section.
- Regarding GO and KEEG analysis, author may highlight the top GO-enriched genes.
Answer:Thank you for your suggestion. In this study, KEGG enrichment chordgram was used to visually display the results of KEGG functional significance enrichment analysis on transcriptome data. We chose each KEGG pathway in significant differences (|log2FC|) of the top 10 gene mapping. And in Figure 3, we highlight theTop enriched genes for KEGG.
- Figure 3, author may highlight the top GO-enriched genes for KEGG.
Answer:Thank you for your suggestion. In Figure 3, KEGG enrichment chordgram was used to visually display the results of KEGG functional significance enrichment analysis on transcriptome data. We chose each KEGG pathway in significant differences (|log2FC|) of the top 10 gene mapping. And in Figure 3, we highlight theTop enriched genes for KEGG.
- Figure 5, author may highlight only common genes (Average data not replicate data). Total identified genes can be provideas a supplementary file.
Answer:Thank you for your suggestion, we have provided the only common genes in Figure 5A and provided the total identified genes as a supplementary file Figure S6.
- Author should make a concluding remarks.
Answer:Thank you for your suggestion, we have made a concluding remarks.
- Basic information (Data) about the transcripts is missing here. Author should provide it as a supplementary file.
Answer:Thank you for your suggestion, we have provided the basic information (Date) about the transcripts as a supplementary file which the excel was named “Statistical table of sequencing data”.
- Author should add the up- and down-regulated genes information in a separate excel file.
Answer:Thank you for your suggestion, we have added the up- and down-regulated genes information in a separate excel file which the excel were named “The DGEs in TR_CT” and “The DGEs in RC_TR”.
- Functional annotation with their enrichment analysis should be provided in a separate excel file.
Answer: Thank you for your suggestion, we have provided the functional annotation with their enrichment analysis in a separate excel file which the excel was named “Statistical table of Kegg functional classification of gene sets”, PDF were named “Histogram of Pathway classification statistics in TR_CT” and “Histogram of Pathway classification statistics in RC_TR”.
- KEGG pathway with their enrichment analysis should be provided in a separate excel file.
Answer:Thank you for your suggestion, we have provided the KEGG pathway with their enrichment analysis in a separate excel file which the excel were named “KEGG enrichment analysis statistical table of TR_CT” and “KEGG enrichment analysis statistical table of RC_TR”.
- Author should provide a basic data file that contains genes information in a separate excel file.
Answer:Thank you for your suggestion, we have provided a basic data file that contains genes information in a separate excel file which the excel was named “leaf genes raw data”.
Reviewer 3 Report
The manuscript entitled “Transcriptome analysis of the responses of rice leaves to chilling and subsequent recovery”, authored by Li and colleagues, deals with the investigation of the chilling resistance mechanisms of rice, combining physiological analysis with transcriptomics ones. The revised manuscript contains truly interesting data that can seriously contribute to the current state of the art. In particular, the manuscript is really well written and structured. However, I would suggest minor changes before it can be judged suitable as a publication in IJMS.
KEYWORDS: Regarding the keywords, they are a useful tool to help indexers and search engines to find relevant papers of interest. If scientific search engines (such as PubMed, Scopus, Google Scholar, etc) can find a potential manuscript by the use of words contained in both title, abstract, and keywords. Consequently, readers will be able to find it too thank this words. An easier search of the manuscript allows to increase the number of people reading your manuscript after publication and, then, to obtain more citations. Consequently, keywords should be words preferably not contained in the title or abstract. This short explanation is to suggest that authors introduce as many keywords as they can (max. 10), and replace those words that are already present at least in the title with new keywords properly related to the reviewed manuscript.
INTRODUCTION: in this section, the authors should introduce some judgment on the possible remedies that currently exist to counteract this problem (solutions derived from plant biotechnology, use of biostimulants, etc.), highlighting possible weaknesses (e.g., many biostimulants exert a beneficial effect through a mechanism of action not yet known=.
RESULTS:
· please, remove 2.3. section. It is not necessary. Or, the authors could decide to merge paragraphs 2.3. - 2.7 into a single paragraph.
· Please, introduce Figure S2 and S3 in the main text, ant not as supplemental material.
MATERIALS AND METHODS:
· Please specify the number of biological and technical replicates used during the experiments, both for the evaluation of morphological, physiological, and transcriptomic traits.
· Section 4.2. should be divided in three different subsection (4.2.1. chlorophylls, 4.2.2. etc), Moreover, the authors should better explain how chlorophylls were quantified. What molar extinction coefficients were used? Why did the authors not provide separate chlorophyll a and b contents? and what about carotenoids?
· The following sentence is not clear “Activities of enzymes were expressed as μg·g−1·FW.”. Proline is not an enzyme.
· The bibliographical references in section 4.2. completely lack.
· The transcriptional analysis part should be better described.
Author Response
- Keywords should be words preferably not contained in the title or abstract. This short explanation is to suggest that authors introduce as many keywords as they can (max. 10), and replace those words that are already present at least in the title with new keywords properly related to the reviewed manuscript.
Answer:Thanks for your suggestions, we have replaced the keywords to new keywords which were related to the reviewed manuscript.
- INTRODUCTION: in this section, the authors should introduce some judgment on the possible remedies that currently exist to counteract this problem (solutions derived from plant biotechnology, use of biostimulants, etc.), highlighting possible weaknesses (e.g., many biostimulants exert a beneficial effect through a mechanism of action not yet known.
Answer:Thanks for your suggestions, we have introduced the biostimulants which are usually sprayed to improve the resistance of rice to low temperature, but the mechanism of the action of biostimulants was still unknown.
- please, remove 2.3. section. It is not necessary. Or, the authors could decide to merge paragraphs 2.3. - 2.7 into a single paragraph.
Answer: Thank you for your suggestion, we have removed the 2.3. section.
- Please, introduce Figure S2 and S3 in the main text, ant not as supplemental material.
Answer: Thank you for your suggestion, we have introduced the Figure S2 and S3 in the main text, and put it in the manuscript.
- Please specify the number of biological and technical replicates used during the experiments, both for the evaluation of morphological, physiological, and transcriptomic traits.
Answer: Thank you for your suggestion, the three biological replicates were used during the experiments, both for the evaluation of morphological, physiological, and transcriptomic traits.
- Section 4.2. should be divided in three different subsection (4.2.1. chlorophylls, 4.2.2. etc), Moreover, the authors should better explain how chlorophylls were quantified. What molar extinction coefficients were used? Why did the authors not provide separate chlorophyll a and b contents? and what about carotenoids?
Answer: Thank you for your suggestion, we have divided the section 4.2 into three different subsection (4.2.1 Determination of chlorophyll content, 4.2.2 Determination of Pro content and 4.2.3 Determination of NSC content). Moreover, we have better explained how chlorophylls were quantified and what molar extinction coefficients were used in manuscript. And we have provided the chlorophyll a, b, a+b and carotenoid contents in supplemental material table S1.
- The following sentence is not clear “Activities of enzymes were expressed as μg·g−1FW.”. Proline is not an enzyme.
Answer: Thank you for your suggestion, we have modified it.
- The bibliographical references in section 4.2. completely lack.
Answer: Thank you for your suggestion, we have added it.
- The transcriptional analysis part should be better described.
Answer: Thank you for your suggestion, we have wrote it more organized way.
Finally, we would like to thank you again for your patience and advice!